# Evaluating a Novel Treatment Adapting a Cognitive Behaviour Therapy Approach for Sexuality Problems after Traumatic Brain Injury: A Single Case Design with Nonconcurrent Multiple Baselines

**DOI:** 10.3390/jcm11123525

**Published:** 2022-06-19

**Authors:** Elinor E. Fraser, Marina G. Downing, Kerrie Haines, Linda Bennett, John Olver, Jennie L. Ponsford

**Affiliations:** 1Turner Institute for Brain and Mental Health, School of Psychological Sciences, Monash University, Clayton, VIC 3800, Australia; marina.downing@monash.edu (M.G.D.); kehaines@bigpond.net.au (K.H.); linda@lindabennettpsychology.com.au (L.B.); jennie.ponsford@monash.edu (J.L.P.); 2Monash-Epworth Rehabilitation Research Centre, Epworth Healthcare, Richmond, VIC 3121, Australia; 3Rehabilitation Medicine, Epworth HealthCare, Richmond, VIC 3121, Australia; john.olver@epworth.org.au

**Keywords:** Sexuality, cognitive behaviour therapy, traumatic brain injury, Rehabilitation

## Abstract

There has been little progress in development of evidence-based interventions to improve sexuality outcomes for individuals with traumatic brain injury (TBI). This study aimed to evaluate the preliminary efficacy of an individualised intervention using a cognitive behaviour therapy (CBT) framework to treat sexuality problems after TBI. A nonconcurrent multiple baseline single-case design with 8-week follow-up and randomisation to multiple baseline lengths (3, 4, or 6 weeks) was repeated across nine participants (five female) with complicated mild–severe TBI (mean age = 46.44 years (SD = 12.67), mean post-traumatic amnesia = 29.14 days (SD = 29.76), mean time post-injury = 6.56 years (median = 2.50 years, SD = 10.11)). Treatment comprised eight weekly, individual sessions, combining behavioural, cognitive, and educational strategies to address diverse sexuality problems. Clinical psychologists adopted a flexible, patient-centred, and goal-orientated approach whilst following a treatment guide and accommodating TBI-related impairments. Target behaviour was subjective ratings of satisfaction with sexuality, measured three times weekly. Secondary outcomes included measures of sexuality, mood, self-esteem, and participation. Goal attainment scaling (GAS) was used to measure personally meaningful goals. Preliminary support was shown for intervention effectiveness, with most cases demonstrating sustained improvements in subjective sexuality satisfaction and GAS goal attainment. Based on the current findings, larger clinical trials are warranted.

## 1. Introduction

Sexuality is a healthy and natural part of living. It is much more than sexual activity and behaviour, encompassing identity, self-esteem, body image, attitudes, motivation, pleasure, and relationships [1,2]. Sexuality is influenced by several sociocultural processes and invariably changes in its meaning and importance across a lifetime [3]. After traumatic brain injury (TBI), a substantial proportion of individuals experience difficulties with sexuality, with prevalence rates generally ranging from 36–54% [4,5,6,7]. The majority report global reductions in sexual arousal and orgasm, perceived importance of sexuality, and frequency of sexual behaviour (i.e., hyposexuality) [6,8,9] In most cases, there is an immediate decrease in sexuality post-TBI with only a small degree of improvement shown across the first year of recovery [10]. Increased sexual arousal and inappropriate sexual behaviour (i.e., hypersexuality) is thought to occur in a smaller proportion of individuals [11]. Hypersexuality, connected to the early post-TBI phase of recovery, is often reversible, whilst persistent hypersexuality is generally underpinned by disinhibition in brain function [12,13].

The biopsychosocial model conceptualises the complex and multifactorial nature of TBI-related sexuality changes as a culmination of biological and medical, psychological and neuropsychological, and social and relationship factors [14]. Neurophysiological mechanisms underpinning sexuality problems may include damage to neuroanatomical regions, altered neurotransmission, and disrupted hormonal regulation [15,16,17]. Although there is agreement that neuroendocrine dysfunction may contribute to post-TBI sexuality changes, including decreased libido, impotence, fertility issues, and irregular menstrual cycles [15,18,19], studies investigating the diagnosis and prognosis of neuroendocrine deficiencies and their implications for sexuality outcomes following TBI are lacking [20]. Although the association between older age and reduced sexuality post-injury has been highlighted in previous research [6,10,21,22,23], the onset of hyposexuality problems following TBI does not appear to be strongly linked to injury severity or time post-injury [6,21,22,24]. With regards to psychological factors, there is strong evidence supporting the role of depression in the onset and maintenance of sexuality problems after TBI [9,22,23]. Self-esteem, anxiety, and antidepressant medication may also be associated with sexuality changes [6,21,22], although it is unclear to what extent concomitant depression also contributes to these associations. Adults with TBI may show increased distractibility, as well as impaired behavioural control, communication, and egocentricity, all of which have the potential to affect one’s capacity to engage in intimate relationships and relate to others [14,15]. Two studies have highlighted an association between decreased social participation and reduced sexuality [23,25], whilst role changes, loss of emotional intimacy, and uninjured partners emotional reactions to the cognitive behavioural changes are likely to pose challenges to sexual readjustment within relationships [26].

Several models and recommendations have been put forth for the management of sexuality problems after TBI. Extant literature advocates for the PLISSIT model [27] as one approach that may be used to address sexual health and wellbeing across TBI healthcare settings [14,28,29,30]. The acronym PLISSIT classifies four levels of intervention: permission to discuss sexuality, provision of limited information, specific suggestions regarding the individual’s sexual problem, and intensive therapy with a qualified healthcare professional. Taylor and Davis [31] later published the extended PLISSIT (Ex-PLISSIT) model which proposes that all levels begin with explicit giving of permission. In previous TBI sexuality research, more emphasis has been placed on addressing the first two levels through the development and evaluation of handouts, booklets, and information resources [32,33,34,35,36,37,38]. As a result, there is little information on interventions for persistent and complex post-TBI sexuality problems at the Specific Suggestions and Intensive Therapy levels. Although previous research has offered broad and nonspecific recommendations for the use of counselling, individual and group psychotherapy, sex therapy, pharmacology, and cognitive and behavioural therapy to address sexuality problems after TBI [5,17,29,32,39,40,41], only a handful of descriptive case reports [41,42,43] and single case studies [44,45] have been completed. Studies have generally focused on treating male sexual dysfunction through the application of standard medical and sex therapy treatments [44,45]. Small sample sizes and a narrow focus on male sexual dysfunction, in addition to an absence of standardised treatment manuals and limited description of how treatment was modified to accommodate TBI-related sequelae, are also relevant limitations of previous research.

To meet the comprehensive and holistic needs of the TBI population, adopting a flexible, integrative, patient-centred, sex-positive, and biopsychosocial approach that emphasises individuals’ strengths rather than limitations is necessary [46,47]. Cognitive behaviour therapy (CBT) is a widely researched, time-limited psychotherapeutic approach that has been shown to be efficacious in the treatment of a wide range of disorders, including TBI-related depression and anxiety [48,49,50] and sleep and fatigue [51,52]. The utility of CBT as a therapeutic option for sexuality disturbances in non-TBI populations is strongly endorsed in the literature [53]. The therapeutic framework proposes that cognitions (thoughts), emotions (feelings), and behaviours all contribute to personal functioning and changes in one domain can lead to changes in others [54]. A mainstay of the CBT approach includes challenging distorted thinking (e.g., negative thoughts related to one’s sexual appeal) and maladaptive behaviours (e.g., avoidance of intimate contact) to achieve more balanced and affirming self-talk and behaviour [55]. Indeed, therapists play an active role in guiding therapeutic interactions and topics of discussion. Relevant to the TBI cohort, the CBT approach acknowledges the multifactorial nature of sexuality and the need to go beyond treating the physiological basis of sexual dysfunction and address psychological and social factors that contribute to sexual wellbeing. Such an intervention can be accessed by adult TBI survivors regardless of sex, gender, sexual orientation, marital status, or type of sexuality issue. Furthermore, CBT includes a significant educational component and can be adapted to accommodate TBI-related cognitive impairment to enhance individuals’ ability to take in and recall information, understand concepts, and remember to complete homework [50,51,52]. Common strategies include greater structure, using more behavioural techniques when cognition is impaired, implementing new skills in vivo, simplifying complex concepts, summarising and repetition of information, pictorial representations of concepts, handouts, external memory aids (e.g., use of a logbook or diary), and provision of organisational support, as well as pre-emptive rest breaks to maintain energy levels [56].

To the best of our knowledge, no study has designed a novel, individualised intervention using a CBT framework tailored to address TBI-related sexuality problems and evaluated its efficacy. The current study aimed to (1) describe a novel, individualised CBT-based intervention in adult TBI survivors with persistent post-injury sexuality problems, (2) use a single case methodology to explore the efficacy of this intervention in improving subjective satisfaction with sexuality, and (3) explore whether this intervention results in improvements in depression, anxiety, self-esteem, social participation, or participants’ attainment of individualised goals.

## 2. Materials and Methods

### 2.1. Study Design

The study used a nonconcurrent, multiple baselines, AB single-case experimental design (SCED) with follow-up (i.e., baseline; treatment; follow-up) repeated across nine participants to explore the effectiveness of an eight-session, individualised CBT intervention on the primary outcome measure of subjective satisfaction with sexuality. Prior to commencing treatment, participants were randomly assigned to baseline durations of 3, 4, or 6 weeks. Experimental control is demonstrated by using the multiple baseline design, which controls for threats to internal validity (e.g., maturation, history) [57,58], whilst the randomisation enhances scientific rigour [59]. The baseline phase was immediately followed by an eight-week treatment phase, before a final eight-week follow-up phase. The Risk of Bias in N of 1 Trials scale (RoBiNT) [60] and the Single-Case Reporting guideline In Behavioural Interventions (SCRIBE) [61] were followed to ensure methodological quality in design and reporting (see Appendix A for scoring).

### 2.2. Participants

Participants were five women and four men recruited via two mechanisms: (1) community advertising to clinicians treating individuals with TBI, and (2) through an established research project that involves collection of follow up outcome data across individuals’ recovery after TBI. Figure 1 shows the flow of participants throughout the study. Informed written consent was obtained from all participants. No financial compensation was provided to the study participants, although participants received treatment free of charge. Inclusion criteria were as follows: (a) aged 18 to 65 years, (b) had sustained complicated mild to very severe TBI, (b) greater than three months post injury, (c) self-reported sexuality disturbance. The following exclusion criteria were used: (a) presence of other neurological disorder, (b) history of psychotic disorder, (c) current alcohol or drug abuse, and (d) insufficient English language or cognitive ability to complete questionnaires or therapy tasks, and (e) sexual dysfunction prior to TBI. Ethical approval for the study was granted by relevant ethics committees. Demographic, injury, and cognitive variables for participants are displayed in Table 1.

### 2.3. Intervention

A detailed treatment guide was developed though an iterative process characterised by a comprehensive review of scientific and grey literature and publicly available information sourced from media, books, expert opinions, and web pages together with discussions and idea generation among a small working group of healthcare practitioners (neuropsychologists, psychologists, doctors) and researchers with expertise in this field. The contents of the treatment guide were organised into 12 modules with accompanying handouts. The overarching aims of CBT were to (1) foster shifts in cognition and/or behaviour that allow individuals/couples to feel more in control of their sexuality, (2) improve satisfaction with sexuality in the individual with TBI, and (3) help individuals with TBI accept and manage changes in sexuality. A medical review was incorporated into the treatment design to aid therapists/clients understanding of the problem and to help differentiate between organic and psychogenic causes and contributing factors.

The intervention consisted of eight 60-min sessions delivered weekly and one booster session completed approximately two months later. The intervention was delivered by two clinical psychologists, KH and LB, licensed to treat clients with CBT and experienced in working with adults with TBI. Therapists had between 13 and 25-years of clinical experience. During periods of non-pandemic related lockdowns, participants had the choice of attending sessions via teleconference (videocall) or in person at the clinicians’ respective private practices. For couples, the intervention was offered to both the participant and their partner.

It was the working group’s intention that the treatment guide be used in a flexible manner. The first treatment session focused on engaging the participant, initiating rapport building, and undertaking a comprehensive assessment and history taking (module 1). The assessment and formulation process followed a biopsychosocial method of classifying predisposing, precipitating, maintaining, and protective factors. The provision of psychoeducation together with goal setting and ongoing rapport building are key features of session 2 that can be built upon and revisited across the duration of treatment. To increase participant engagement, generating the clinical formulation and defining specific goals was intended to be a collaborative process.

For sessions 3–7, therapists were encouraged to apply clinical judgement in identifying, selecting, and modifying the delivery of modules to suit the needs of the client, taking into consideration their presenting problem, clinical formulation, and goals. Hence, the content of each session and the overall number of modules delivered was expected to vary according to the individual/couple. As a tangible representation of each module, the purpose of the accompanying handouts was to provide structure to the treatment, as well as aid communication and delivery of information between the therapist and client. Handouts were used not only in the context of building psychoeducation but also the exploration of cognitive and behavioural strategies.

The purpose of Session 8 was consolidation of treatment content, skills, and strategies. Functional goals were reviewed as a marker of treatment progress, whilst relapse prevention was explored in the context of supporting maintenance of treatment gains. When delivering the treatment, clinical psychologists implemented several strategies to support engagement in sessions and retention of information. Key strategies included educational scaffolding, visual handouts, written summaries, repetition, and simplification of concepts. The degree to which strategies were applied, however, varied between individuals according to their needs. The treatment structure as it related to sessions and delivery of modules and accompanying handouts is displayed in Table 2.

### 2.4. Measures

#### 2.4.1. Measures of Participant Baseline Characteristics

The National Adult Reading Test (NART) [62] was used as an estimate of premorbid intellectual ability in the current study. New learning and memory were assessed using the total words recalled in trials 1–5 on the Rey Auditory Verbal Learning Test [63], whilst executive function and speed of information processing were measured using the total time taken to complete the Trail Making Test Part B [64] and digit symbol-coding test [65], respectively. The Fatigue Severity Scale (FSS) [66] together with the pain and independence subscales of the Traumatic Brain Injury-Quality of Life (TBI-QOL) [67] were also administered (See Appendix A for scores).

#### 2.4.2. Primary Outcome Measure

For the purposes of this study, the authors developed a rating scale to measure participants’ subjective satisfaction with their sexuality [68]. Participants were asked to rate the following question ‘How satisfied are you with your current sexuality?’ on a 7-point Likert scale ranging from ‘extremely unsatisfied’ to ‘extremely satisfied’.

#### 2.4.3. Secondary Outcome Measures

Several secondary measures were used to provide converging evidence for treatment effectiveness. The Brain Injury Questionnaire of Sexuality (BIQS) [69] is a 15-item, self-report questionnaire comprising three subscales measuring sexual functioning, relationship quality and self-esteem, and mood. Respondents are required to compare aspects of their sexuality with preinjury status on a 5-point Likert scale ranging from “greatly decreased” to “greatly increased”. Total sexuality scores between 14 and 44 are classified as decreased from pre-injury levels, the same for scores of 45, and increased from pre-injury levels for scores between 46 and 75.

The Hospital Anxiety and Depression Scale (HADS) [70] was used as a reliable 14-item measure of anxiety and depression symptoms. Higher scores on the HADS subscales reflect higher levels of depression and anxiety. The Rosenberg self-esteem scale (RSES) [71] was utilised as a 10-item measure of self-esteem. On this measure, higher scores reflect better self-esteem, with scores less than 15 suggestive of low self-esteem. As a reliable and valid measure of social participation after TBI, the Participation Assessment with Recombined Tools-Objective (PART-O) [72] comprises 17 items across three subscales measuring productivity, social relations, and ‘out and about’ (e.g., going to the movies). The averaged total score was used as an indication of overall social participation, with higher scores indicative of greater social participation.

Goal attainment scaling (GAS) [73] was used to set individualised goals, allowing for measurement of personally meaningful progress [60]. Possible outcomes were defined according to a standard five-point symmetrical scale (−2 a lot less than expected; −1 less than expected; 0 at expectation; +1 more than expected; +2 a lot more than expected). Level of goal attainment was assessed at post-intervention and follow-up timepoints. GAS goals were initially set at −1 to allow for measurement of deterioration [74].

### 2.5. Treatment Integrity

Treatment sessions were audio recorded for assessment of treatment integrity. Specifically designed treatment integrity monitoring forms were developed to measure, (1) overall adherence to elements common in a CBT approach, (2) adherence to the chosen module(s) used in the session, and (3) therapist competency in module delivery [52]. The three domains were rated on an 8-point Likert scale ranging from ‘unacceptable’ to ‘excellent’. Two randomly selected recordings per participant were chosen for evaluation by an independent practitioner with 21 years of professional clinical psychology experience.

### 2.6. Procedure

Data were collected between March 2021 and March 2022. Participants were provided with information about the research project and screened for eligibility via telephone. Participants who met eligibility criteria were scheduled for an initial baseline assessment. Participants were then randomly allocated to a baseline monitoring phase of 3, 4, or 6 weeks. The target behaviour (i.e., subjective satisfaction with sexuality) was assessed three times per week via a text message reminder system. Ratings were recorded online across all three study phases (baseline, intervention, follow-up) to provide a continuous measure of rate of change. Secondary outcome measures assessing sexuality, depression and anxiety, self-esteem, and social participation were collected prior to the intervention (conclusion of the baseline period), at the conclusion of the intervention, and eight weeks post conclusion of the intervention. The majority completed secondary measures on the online platform Qualtrics^TM^ (Qualtrics, Provo, UT, USA) to minimise bias in data collection. They were otherwise collected by a researcher independent of the therapist. Goal attainment scaling goals were set in collaboration with the therapist and reassessed at two timepoints (at the conclusion of the intervention and eight weeks post conclusion of the intervention). Participants received eight one-hour therapy sessions with clinical psychologists KH or LB either face to face or via teleconference (videocall). After cessation of therapy, a 30–45-min semi-structured interview was administered by the first author to evaluate the usefulness of the intervention, to identify its strengths and limitations and ascertain whether the participant would recommend it to others in the future.

### 2.7. Data Analysis

Descriptive statistics were generated for all variables of interest. For cognitive measures, raw scores were converted to standardised z-scores using age- and education-based normative data. Primary outcome sexuality satisfaction ratings were displayed graphically using Graphpad Prism (Version 8) and evaluated using visual analysis in line with established guidelines proposed by Lane and Gast [75], incorporating both within and between phase analysis. The SCDA plugin for R [76] was used to fit a linear trend to data using the split middle method [59,77]. Stability was defined as 80% of data points being within an envelope of ±25% of the phase median. When evidence for a functional effect was present, we proceeded to estimate the effect sizes using statistical analysis.

Statistical analyses used the Tau-U statistic, which is a nonparametric index of the percentage of the data that do not overlap minus the percentage of the data that overlap between phases [78]. Tau-U is a distribution-free non-parametric technique, with an index well-suited for small datasets and is useful in aggregating data across phases to provide an overall effect size [78]. Baseline correction was applied when Tau-U values for baseline exceeded 0.20 and the trend occurred in the same direction as the aims of the intervention (i.e., when baseline displayed a trend to improving sexuality satisfaction) [79]. Tau-U values below 0.2 were considered small, 0.2–0.6 medium, 0.6–0.8 large, and greater than 0.8 large to very large [80]. Analysis was conducted using the online calculator at http://www.singlecaseresearch.org/calculators/tau-u (accessed on 15 January 2022).

Informal comparisons were made between pre-treatment, post-treatment, and follow-up secondary outcome measures. Functional change on participants’ individual GAS goals were descriptively explored with any change considered to constitute clinical significance [81]. Independently collected semi-structured interview data related to participants’ experience of the intervention were also qualitatively examined, although they will be reported in later publications.

## 3. Results

### 3.1. Case Description

Nine community dwelling adults with a diagnosis of TBI were recruited into the study (five female, four male). Participants varied in age (31–64 years), severity of injury as measured by PTA duration (<1–70 days), and time since injury (0.90–33 years), and most displayed impairment on at least one measure of cognition. Five participants were receiving psychological intervention for other issues, unrelated to sexuality, at the time of study participation. The majority completed the treatment via teleconference (videocall), with only one participant (AA) attending sessions in person. Participant DD withdrew during the treatment phase due to extenuating circumstances that meant they did not have the time or capacity to continue with the study. All other participants completed the study in its entirety. No adverse events were recorded for any participant across the duration of the study.

With regards to presenting problems in the current sample, all participants presented with sexuality issues consistent with reduced sexuality, or hyposexuality, regardless of age, sex, marital status, or time post injury. Individuals endorsed reductions in sexual desire and arousal and ability to climax (AA, BB, CC, DD, EE, FF, GG, HH, II), self-esteem (DD, HH), body image (BB, DD, II), and communication (FF, GG, II). Of the four males, three (AA, CC, EE) presented with erectile dysfunction, and for those who had partners, there was a consistent reduction in frequency of, and satisfaction with, sexual intimacy within the relationship (CC, DD, EE, HH, II). Despite overlap in the nature of participants’ sexuality complaints, establishing the probable aetiological basis of the individual’s issue through comprehensive assessment and formulation was a crucial first step of the intervention. For two participants (AA, FF), this also involved undergoing medical review to clarify the relative contributions of injury-related, neurological, and/or biological factors.

Given the diversity in presenting problems, it was of the utmost importance that treatment planning and delivery were individualised and tailored. To illustrate how this was implemented in this study, it can be helpful to briefly describe and compare cases with similar characteristics. CC and EE were both males in their 60′s and married, and both reported a loss of intimacy in their relationship related to the post-injury onset of erectile dysfunction. For CC, treatment involved increasing his awareness of factors related to erectile dysfunction and trialling psychosexual skill exercises. By the final session, CC was able to successfully obtain an erection. On the other hand, EE’s erectile dysfunction was a direct result of damage sustained to his pelvis and genital area in the accident. As a result, treatment focused on expanding EE’s understanding of sexuality and masculinity and working on ways in which EE could facilitate emotional closeness and connectedness’ with his wife outside of sexual intercourse. Single and actively dating, AA was the third participant who presented with erectile dysfunction. In this case, low self-esteem, high levels of performance anxiety, and lifestyle factors (weight, alcohol, smoking) were key contributing factors. Among other things, the therapist worked with AA on defining his personal strengths, recognising that sexual intercourse is only one part of sexuality, and specifying what has facilitated and inhibited desire and arousal for AA in the past. Cognitive impairment combined with unrealistic treatment expectations, however, limited his ability to engage in CBT techniques aimed at modifying thoughts, challenging unrealistic standards regarding sexual function, and engaging in behavioural experiments.

For female participants, there was equal variability in factors contributing to reduced sexual drive, desire, and arousal. FF experienced changes in sensations, which meant she was unable to tolerate touch. Trialling remedial massage in a non-intimate context was one behavioural experiment that FF and the therapist subsequently worked on. Negative body image formed part of the clinical formulation for three participants’ (BB, DD, and II), although treatment needed to be tailored to address other relevant factors including anticipatory anxiety to chronic fatigue relapses (BB), high levels of self-blame (DD) and irritability and anger outbursts (II).

Only two participants, II and HH, completed the treatment with their partner. The latter was a unique case in that HH’s partner had taken on a carer’s role in his recovery, and at one-year post-injury, it was his partner who was reporting low sexual desire and arousal, whilst HH himself struggled with low self-esteem and difficulties communicating his sexual needs. In this case, a core facet of the treatment was delineating intimacy and sexuality from sexual intercourse and enhancing the couple’s comfort and communication on the topic. Overall, the case descriptions offer insight into the complexities of the presentations and treatment of sexuality after TBI, highlighting the need to adopt a flexible and individualised approach predicated on comprehensive assessment and formulation.

### 3.2. Treatment Adherence and Integrity

Completed by a clinical psychologist, integrity monitoring of the delivery of the treatment indicated ‘acceptable’ to ‘excellent’ ratings for overall delivery (mean 5.76 [range 4.5–7.5]), adherence (mean 6.24 [range 4.5–8]), and competency (mean 5.55 [range 4–7]). An overview of each participant’s target behaviour (sexuality satisfaction, with improvements reflected by higher scores) is presented in text and in Figure 2.

### 3.3. Self-Reported Subjective Sexuality Satisfaction

The application of the stability envelope suggested stability of baseline phase for participants AA, EE, FF, and II. Participants GG and HH demonstrated decreasing baseline trend, whilst participants BB, CC, and DD showed increasing baseline trend. For the latter participants, a visual trend correction using the split middle method was applied. Participants AA, CC, FF, GG, HH, and II showed an increasing trend in the therapeutic direction following the introduction of the intervention. Therapeutic gains were maintained between treatment and follow-up phases for participants CC, FF, HH, and II, whilst a decreasing trend was shown in follow-up phase data for participants AA and GG. Although participant BB and EE showed no significant trend change in sexuality satisfaction during the treatment phase, an increasing trend was shown during the follow-up phase suggestive of the delayed onset of therapeutic gains following the delivery of the intervention.

With respect to the between phase change, the introduction of treatment was associated with a median level increase in subjective sexuality satisfaction for participants AA, CC, FF, GG, and II. Participant HH demonstrated a marginal between phase increase in sexuality satisfaction, whilst the introduction of the intervention was associated with a delayed median level change in sexuality satisfaction for participants BB and EE, occurring during the follow-up phase of the study.

Tau-U analysis was used to determine statistically significant changes in subjective satisfaction with sexuality between baseline and treatment phases and treatment and follow-up phases. Sexuality satisfaction significantly increased between the baseline and treatment phases for participants AA, CC, FF, GG, and II. Tau-U analyses further demonstrated statistically significant increases in subjective sexuality satisfaction between treatment and follow-up phases for participants AA, BB, CC, EE, HH, and II (See Table 3).

### 3.4. Secondary Outcome Measures

Descriptive analyses of the BIQS, HADS-A, HADS-D, RSES, and PART-O revealed variable results (see Table 4). On the BIQS, all participants’ pre-treatment total sexuality scores were classified as decreased from pre-injury sexuality status. All participants displayed increased sexuality scores at post-treatment. The participants who commenced treatment with elevated symptoms of anxiety (AA, FF, GG) and depression (FF, GG) recorded no meaningful change on the HADS at post-treatment. With regards to self-esteem, however, those who were classified as having low self-esteem at pre-treatment (AA and HH) demonstrated significantly improved self-esteem at post-treatment, which was maintained at follow-up. Social participation measured by the PART-O was considered a more distal secondary outcome measure and did not show any convincing change between pre-treatment, post-treatment, and follow-up measurements. 

All participants reported improvement in at least one goal area following treatment, which was generally maintained, and even showed improvement for participants BB, EE, HH, and II when rated at the conclusion of the eight-week follow-up period. Although participant AA reported the attainment of goals following the intervention, this change was not maintained for two of the goals at follow-up (see Table 5).

## 4. Discussion

This study served to evaluate the preliminary efficacy of a novel intervention using a CBT framework that aimed to improve individuals’ satisfaction with sexuality following TBI. Hyposexuality problems identified in this study included erectile dysfunction, reduced sex drive and orgasm, negative changes in self-esteem and body image, loss of relationship intimacy, and difficulty establishing new intimate partnerships. Five participants showed treatment response in the therapeutic direction following the introduction of the intervention, whilst an additional three participants demonstrated delayed treatment response. Gains in sexuality satisfaction were generally maintained at two months following the completion of treatment. The finding that participants varied in their response to treatment was not unexpected given the individualised nature of the intervention. Signals of efficacy were also identified on secondary outcome measures of sexuality, functional goal attainment, and self-esteem. The intervention demonstrated good feasibility and adequate treatment adherence. Although the findings are encouraging, it is of note that only three participants recorded feeling ‘satisfied’ with their sexuality on the primary outcome measure at post-treatment and four at follow-up timepoints. Furthermore, one participant demonstrated the limited maintenance of treatment gains on the primary outcome measure and functional goals. Factors that may have contributed to this included the participants’ expectation that the treatment would resolve all physical sexual issues, as well as ongoing pandemic-related lockdowns that prevented socialisation and dating.

The comparison of results with previous research is limited by the lack of studies that have previously reported on the treatment of persistent or complex post-TBI sexuality problems, with none having taken a CBT approach but rather delivered standard sex therapy and medical techniques [44,45]. Certainly, the current findings align with recommendations regarding the need for evidence-based interventions to be accessible and available at the Intensive Therapy level of the PLISSIT/Ex-PLISSIT model [14]. The proposition that intensive therapy may be offered at any stage of recovery is supported by the current research [31]. Indeed, the primary barrier to implementing intensive therapy is the lack of evidence regarding what specific techniques and tools are efficacious in ameliorating persistent post-TBI sexuality disturbances and how treatments should be tailored to suit the needs of individuals and couples following TBI [7,17]. The case descriptions outlined in this research demonstrate the complexity and nuance that needs to be considered in sexuality assessment and treatment after TBI. The continued development of interventions that are purposefully designed to meet the needs of individuals and couples following TBI is necessary to facilitate the clinical implementation of structured approaches to sexuality management, such as the PLISSIT/Ex-PLISSIT model.

### 4.1. Clinical Implications and Future Research

This study offers provisional support for the development of an evidence-based adapted CBT therapeutic intervention for people experiencing persistent and complex sexuality difficulties after TBI. The findings have important clinical implications for the treatment of sexuality in a TBI population. Given the heterogeneous nature of TBI, any one or combination of factors may lead to sexuality changes after TBI. Medical issues, TBI sequelae, pain, chronicity of TBI, profile of cognitive impairment, marital status, length of relationship, age, and lifestyle factors were all intricately interwoven with the sexuality problems addressed in this study. As such, adopting a holistic, flexible, and patient-centred approach is necessary to ensure tailored treatment planning and delivery.

Beyond consideration of the multifactorial aetiology of sexuality changes after TBI, there are implications associated with using a CBT framework delivered by a clinical psychologist that are worth highlighting. Specifically, it is important to note that in cases where there has been permanent alteration in an individual’s physiological sexual response, an eight-week CBT intervention is unlikely to improve physical sexual function. At best, treatment will help to facilitate the emotional acceptance of the primary dysfunction, a greater understanding of sexuality, and ways to facilitate intimacy in ways other than sexual intercourse. Furthermore, there are cases where sexuality changes may directly reflect injury-related physical limitations, such as pain, increased or decreased sensitivity, and muscle weakness or spasticity, which influence body positioning and movement. In these instances, a psychological-based intervention characterised by eight sessions with a clinical psychologist may not be beneficial, rather a different discipline, such as physiotherapy or occupational therapy, may be better placed to address the issues. Indeed, there is a strong consensus that interdisciplinary sexuality service delivery is required to meet the needs of the TBI population [17,29,30].

Although further studies are required, this research provides a model for how sexuality changes may be addressed in this clinical population. Future studies evaluating sexuality treatment for individuals with TBI may benefit from larger or targeted sampling, which may allow for the evaluation of the extent to which the nature of the presenting sexuality issues or factors, such as cognitive impairment, cultural background, age, sex, gender, sexual orientation, marital status, duration of time post-injury, and other biopsychosocial variables influence responses to treatment. Increasing understanding of the potential impact of such variables will assist the design and implementation of interventions and further increase clinicians’ awareness of factors to consider when addressing TBI-related sexuality changes. Additional research is needed to identify the optimal number and length of sessions and support the development and dissemination of the treatment guide and associated resources. Finally, there is a need for greater recognition and inclusion of participants with diverse gender and sexuality backgrounds, including those who identify as LGBTIQ+, in future research.

### 4.2. Limitations

There are several limitations to this study. First, the small number of participants included in this study limits the interpretation of the results. Participants had also undergone rehabilitation in the context of a no-fault accident compensation system, which may limit the generalisability of study findings to individuals with TBI who have not received rehabilitation. In addition to the sample consisting of white cisgender, heterosexual participants, it is worth noting the lack of recruited participants aged between 18 and 30 years, which may represent a unique period in the life cycle for sexual health and wellbeing. The results may also not extrapolate to individuals with TBI from non-Western countries or cultural backgrounds, especially those with more conservative attitudes towards sexuality. Although the diversity in sexuality problems highlighted in the current research is likely indicative of that seen in the community, the lack of uniformity prevents comparisons between individuals in terms of level and rate of improvement. Furthermore, it is unclear which components of the intervention influenced therapy response, given that an individualised approach was adopted, and a variety of therapeutic techniques were utilised. Importantly, however, the single-case experimental design did allow for the evaluation of a tailored intervention in a heterogeneous group of adults with TBI at an individual level. Other limitations, such as the lack of participants and therapist blinding and potential measurement error, also apply. GAS ratings may be subject to Rater bias as they were completed by non-blinded participants in cooperation with therapists and, therefore, should be interpreted cautiously. The authors acknowledge that the primary outcome measure was not psychometrically validated, however, there were no other validated measures that would be considered appropriate to use in this specific study. Indeed, there continues to be a lack of precision in TBI sexuality measurement, evidenced by the variability in measures utilised in the few treatment studies completed to date [44,45]. Another limitation to this intervention is the lack of involvement from occupational therapists and physiotherapists in the assessment, as well as treatment planning and delivery process. Finally, a longer follow-up (e.g., 6 to 12 months) would have been valuable to see whether improvements on outcome measures were maintained in the longer term.

## 5. Conclusions

To the best of our knowledge, this is the first published account of a preliminary empirical evaluation of a novel intervention adapting a CBT approach to treat sexuality problems after TBI. The findings make a unique contribution to the sexuality and TBI literature and confirm that using a CBT framework to treat TBI-related sexuality difficulties may represent a promising therapeutic avenue. As a rigorous, albeit small-scale, sexuality intervention study of individuals with TBI, this research provides preliminary evidence of efficacy and feasibility and thus is relevant to stakeholders in research, policy, and clinical practice.

## Figures and Tables

**Figure 1 jcm-11-03525-f001:**
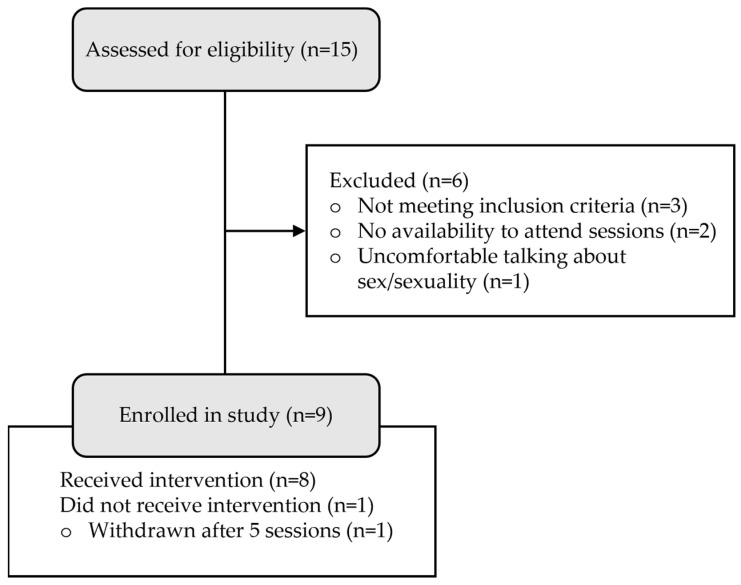
Recruitment and flow of participants throughout the study.

**Figure 2 jcm-11-03525-f002:**
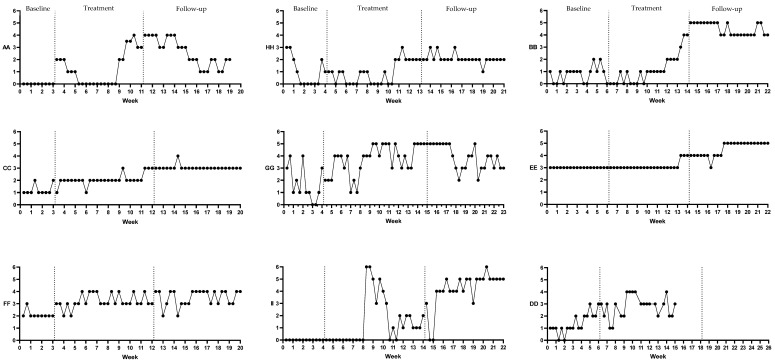
Participants’ self-reported subjective sexuality satisfaction across baseline, intervention, and follow up phases.

**Table 1 jcm-11-03525-t001:** Demographic, injury, and cognitive characteristics of the participants.

Variables	Participant								
	AA	BB	CC	DD	EE	FF	GG	HH *	II *
**Demographic** **data**									
Age (years)	49	31	64	47	63	56	33	33	41
Sex	Male	Female	Male	Female	Male	Female	Female	Male	Female
Education (years)	12	18	14	12	12	15	16	14	18
Preinjury relationship	Single	De-facto	Married	Married	Married	De-facto	De-facto	De-facto	Married
Baseline relationship	Single	Single	Married	Married	Married	Single	Single	De-facto	Married
**Injury characteristics**									
Cause of TBI	Bicycle Accident	MVA	Work-related injury	MVA	MVA	MVA	MVA	Fall	Pedestrian struck by car
PTA duration (days)	70	16	21	1	0.5	20	8	68	23
Worst GCS score	3	15	8	^b^	15	14	3	3	13
Time since injury (years)	33	6	6	5	2	3	1	1	0.90
CT brain imaging	Petecchial haemorrhages in frontal and paraventricular regions	Frontal contusion, subarachnoid haemorrhage, contrecoup injury with contusion in the orbitofrontal region	NAD	Fractured occipital lobe, epidural haematoma, subarachnoid haemorrhage pneumocephalus, thrombosis of transverse sigmoid sinus and jugular vein	Interhemispheric subarachnoid haemorrhage	NAD	Fractured parietal and temporal bones, contusion and intracerebral haemorrhage in temporal region, extra-axial haemorrhage	Fractured occipital lobe, subfalcine herniation, inferior frontal contusion, grey and white matter loss, haemorrhage	Left subdural haematoma, contusions in right frontal and temporal regions, fractured left temporal bone
**Cognitive performance**									
Digit Symbol Coding Test z-score	−1.33	0.00	−1.00	−1.00	−1.67	0.00	NA	NA	0.67
Verbal Learning RAVLT Trials 1–5 Sum z-score	−1.52	−0.92	−0.70	0.22	−0.96	1.53	NA	−2.12	0.22
Trail Making Test Part-B z-score	−0.67	−0.67	−1.00	−0.67	−1.00	0.00	NA	NA	1.33
Estimated Premorbid Intellectual Functioning (NART Standard Score)	103	104	105	113	115	117	NA	NA	100

TBI, traumatic brain injury; MVA, motor vehicle accident; PTA, post traumatic amnesia; GCS, Glasgow Coma Scale; CT, computed tomography; NAD, no abnormality detected; NA, not applicable—no formal testing was able to be undertaken; DSCT, digit symbol coding test; RAVLT; Rey Auditory Verbal Learning Test; NART, National Adult Reading Test. ^b^ Blank cells represent missing data. * Completed the intervention with their partner.

**Table 2 jcm-11-03525-t002:** Treatment structure and modules.

Session	Module	Objective
1–2	Module 1: Assessment and Formulation	Work closely with the individual/couple to develop a shared understanding of the problem and collaboratively set treatment goals.
	Module 2: Psychoeducation and Goals	Explore the individual’s/couple’s understanding of sexuality and provide psychoeducation surrounding sexuality, TBI, and overall health and wellbeing.
3–7	Module 3: Self-esteem and Body Image	Explore self-esteem or body image and identify and adjust biased expectations, negative self-evaluations, and rules and assumptions
	Module 4: Understanding Arousal	Define desire and arousal, explore dimensions of touch, identify brakes and accelerators
	Module 5: Reframing thoughts	Explore links between thoughts, emotions, and behaviours. Employ cognitive restructuring techniques that allow the individual/couple to feel in control of their sexuality.
	Module 6: Communication	Understand expressive and receptive listening skills using modelling and in-session coaching.
	Module 7: Relaxation	Teach mindfulness, breathing, and progressive muscle relaxation techniques to enhance sexual experiences
	Module 8: Psychosexual Skill Exercises	Practise exercises targeted to the individual’s/couple’s sexual problem.
	Module 9: Social Skills	Establish relevant techniques to increase positive and rewarding social behaviours and decrease negative social behaviour.
	Module 10: Sleep and Fatigue Management	Identify practical strategies to improve sleep and manage physical and mental fatigue.
	Module 11: Medical Review	Assess neurological/medical basis to sexuality problem.
8	Module 12: Relapse Prevention	Summarise skills and content learned throughout treatment and generate plan for managing setbacks.

TBI, traumatic brain injury.

**Table 3 jcm-11-03525-t003:** Tau-U planned comparison for self-reported ratings of sexuality satisfaction.

Participant	Baseline Corrected	Baseline vs. Treatment	*p* Value	Treatment vs. Follow-up	*p* Value
AA	No	0.54 *	0.018	0.54 **	0.001
BB	No	0.13	0.461	0.96 **	0.001
CC	No	0.74 **	0.001	0.86 **	0.001
DD	^b^	^b^	^b^	^b^	^b^
EE	No	0.22	0.211	0.89 **	0.001
FF	No	0.84 **	0.001	0.29	0.084
GG	No	0.69 **	0.001	0.02	0.923
HH	No	0.01	0.980	0.68 **	0.001
II	No	0.53 *	0.008	0.65 **	0.001

* Significance at *p* < 0.05; ** Significance at *p* ≤ 0.001. ^b^ Blank cells represent missing data due to participant withdrawal.

**Table 4 jcm-11-03525-t004:** Participants’ pre-treatment, post-treatment, and follow-up raw scores for secondary outcome questionnaire measures.

Participant	BIQS			HADS-A			HADS-D			RSES			PART-O		
	Pre	Post	Follow-up	Pre	Post	Follow-up	Pre	Post	Follow-up	Pre	Post	Follow-up	Pre	Post	Follow-up
AA	29	44	39	9	8	5	1	2	8	15	18	16	1.78	1.95	1.98
BB	34	49	46	4	1	1	2	2	3	28	30	30	1.38	1.68	1.66
CC	26	35	41	2	1	1	3	0	1	20	20	22	2.11	1.61	2.04
DD	21	^b^	^b^	10	^b^	^b^	11	^b^	^b^	9	^b^	^b^	2.42	^b^	^b^
EE	22	50	28	0	1	1	2	3	3	21	26	26	0.84	1.25	1.30
FF	29	33	24	14	11	12	10	8	10	21	22	23	1.48	1.21	1.42
GG	22	39	34	13	8	13	13	9	8	17	18	11	1.35	1.46	1.51
HH	37	40	41	6	5	5	7	10	9	10	15	17	1.79	1.79	1.90
II	19	25	20	5	4	3	2	0	0	18	19	20	2.84	2.94	3.03

BIQS; Brain Injury Questionnaire of Sexuality; HADS-A, anxiety subscale from the Hospital Anxiety and Depression Scale; HADS-D, depression subscale from the Hospital Anxiety and Depression Scale; RSES, Rosenberg Self-Esteem Scale; PART-O, Participation Assessment with Recombined Tools-Objective. Note: Data represents participants’ raw scores for each measure. ^b^ Blank cells represent missing data due to participant withdrawal.

**Table 5 jcm-11-03525-t005:** Description and attainment of participant GAS goals.

Participant	Goals	Pre-Treatment	Post-Treatment	Follow-up
AA	1. To work towards having an emotionally connected/supportive relationship with a female	−1	0	−1
	2. To work on erectile dysfunction	−1	+1	0
	3. To learn how to communicate clearly and honestly before engaging in sexual activity	−1	0	−1
BB	1. To feel better about body image	−1	+2	+2
	2. To not feel anxious about experiencing chronic fatigue syndrome relapses	−1	+1	+2
CC	1. To be physically intimate with partner	−1	0	0
	2. To work on erectile dysfunction	−1	0	0
DD	1. To be physically intimate with partner	−1	^b^	^b^
	2. To improve self-esteem	−1	^b^	^b^
EE	1. To increase understanding of sexuality and TBI	−1	+2	+2
	2. To explore pain management options	−1	+1	+2
	3. To explore masculinity in the context of pre- vs. post-injury self	−1	+2	+2
FF	1. To be confident to put self out there	−1	−1	0
	2. To feel more in touch with body	−1	0	0
GG	1. To improve desire/motivation/sex drive	−1	+1	+2
	2. To work on communication and reduce emotionally closing off	−1	+1	+1
HH	1. To improve sexual relationship with partner	−1	+2	+2
	2. To feel better about sexuality	−1	+1	+1
	3. To improve self-esteem	−1	+1	+1
	4. To reduce the need for partner to take on a care-taking role	−1	+1	+2
	5. To increase understanding of sexuality and TBI	−1	+2	+2
	6. To know more about partner’s sexuality	−1	+2	+2
II	1. To increase knowledge of how TBI has impacted sexuality	−1	+2	+2
	2. To improve management of fatigue	−1	+1	+2
	3. To create more opportunities for intimacy behaviours	−1	+1	+1
	4. To develop more effective communication skills	−1	+2	+2

GAS, goal attainment scaling; TBI, traumatic brain injury. ^b^ Blank cells represent missing data due to participant withdrawal.

## Data Availability

The data presented in this study are available on request from the corresponding author. The data are not publicly available due to ethical restrictions.

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
