# Peer review of "Evaluating a Novel Treatment Adapting a Cognitive Behaviour Therapy Approach for Sexuality Problems after Traumatic Brain Injury: A Single Case Design with Nonconcurrent Multiple Baselines"

_jcm, 2022, doi:10.3390/jcm11123525_

Round 1

Reviewer 1 Report

In a submission to the Journal of Clinical medicine entitled, "Evaluating a novel treatment adapting a cognitive behaviour 2 therapy approach for sexuality problems after traumatic brain injury: A single case design with nonconcurrent multiple baselines ", the authors evaluate the efficacy of rehabilitative intervention using a cognitive behavior therapy (CBT) to treat sexuality problems post-TBI. The authors show that CBT-induced sustained improvements in subjective sexuality satisfaction and GAS goal attainment in patients with TBI. The study is well conducted and the results are interesting. Diverse factors could explain the impact of sex on TBI symptoms and recovery, including hormonal contributions. There are few reports of menstrual cycle fluctuations in females after TBI. I would recommend the authors add available (if any) information, where applicable, on the impact of circulating female sex hormones on sexuality and recovery after injury.

Reviewer 2 Report

The issues addressed by this preliminary study of single cases are of absolute interest due to the close relationship between sexuality and the overall well-being of the person, even after the often destructuring experience of a head injury. One merit of this research was that it was not limited to the description of the broad symptomatology of the sexual sphere, as happens in many papers on this topic, but that it proposed an approach based on a cognitive behavioral therapy (CBT) framework, identified as therapeutic tool capable of acting on the three levels of disability in the superior cortical functions of these patients (cognitions, behaviors and emotions). The adoption of a flexible, patient-centered and goal-oriented approach has finally made it possible to examine the organic factors that often negatively interfere with sexual activity in TBI patients. The study design adopted seems to represent the great variability of the conditions clinics of the small sample of subjects recruited. The final section (4.2 Limitations) is honest and allows, along with what is detailed in 4.1. (Clinical Implications and Future Research) to identify future and desired lines of research.

 To further improve the quality of the paper I suggest some points:

1 1-     In "Introduction" mention the possibility that after TBI there may also be aspects of hypersexuality even if less frequent, often reversible and connected to the early post TBI phase (Cluver-Bucy syndrome), and, if later, in part secondary to a generalized disinhibition of brain function. (see for example: Britton K. R. (1998). Medroxyprogesterone in the treatment of aggressive hypersexual behavior in traumatic brain injury. Brain injury, 12 (8), 703–707. https://doi.org/10.1080/026990598122269; Komisaruk , B. R., & Rodriguez Del Cerro, M. C. (2015). Human sexual behavior related to pathology and activity of the brain. Handbook of clinical neurology, 130, 109–119. Https://doi.org/10.1016/B978-0- 444-63247-0.00006-7.)

2 2-      In 2.1- "Study Design", line 116 (or in Discussion) explain the usefulness of a baseline of variable duration (3,4,6 weeks) since the large interval from the TBI could in itself be a guarantee of stability of the behavior under study.

3 3-      In 2.2 “Partecipants” describe any reasons that limited the sample size to 9 subjects and if the number of excluded subjects was recorded, with the relative motivation.

4 4-      In 2.3 "Secondary Outcome Measures" confirm the appropriateness of a scale, which I do not know personally: The Hospital Anxiety and Depression Scale (HADS) (line 192) which seems to refer to hospitalized patients, while the subjects of this paper are out -patients

5 5-      In 4.2 "Limitations" I would also point out the lack of recruited subjects between the ages of 18 and 30, a period of the life cycle that is certainly also important for sexual life and, perhaps, in a different way than what happens at a higher age.

Reviewer 3 Report

Although the article is interesting, there are several aspects to improve in it.

Keywords should be checked for example neurosexuality if it is a MESH keyword.

The biopsychosocial model is assumed to be the approach to be used. I think there are specific intervention models in sexuality such as EX-PLISSIT

Line 45-52 It should be expanded and structured

Line 53 What are several models and frameworks? The authors do not specify. In general, these paragraphs are somewhat hard to read, they do not follow a common thread.

The objectives can be improved in writing and in SMART formulation. In addition, the authors speak of approaching using a holistic approach, which is the basis of a discipline not addressed by the authors, Occupational Therapy.

Participants

I have doubts regarding the exclusion criteria, the authors indicate "cognitive capacity to provide informed consent". This makes me doubt... if there is cognitive involvement... I don't know to what extent this study is ethical. Nor do they include anything about legal incapacity or the like... if it is a disabled person, it should not be included in the study. I believe that there is a very fine ethical line in this aspect that the authors should describe more exhaustively.

Table 1 => In the field of acquired disability and sexuality, there are fundamental aspects such as whether the person had a partner prior to the injury or not, whether they had previous sexual experiences. I also don't understand the case of AA, which states that he is 33 years old without injury and is 49, which leads us to the fact that he was 16 in the bicycle accident... he still had no previous sexual experiences. Previous sexual education received, sexual roles or sexual interests are not addressed either. I don't understand the last case either 0.9 time since injury in years..... The authors assess at the cognitive level... but there is no assessment of other necessary parameters such as advanced social skills, cognitive flexibility or motor or processing skills such as planning or sequencing activities and tasks...

Point 2.3 is not well described, I still have many things to understand how they were executed and carried out. There is even no physiotherapist or occupational therapist during the intervention. Even a figure that describes the process would be interesting. The authors talk about the content of the guide, and here they do talk about social skills but they have not been evaluated, nor important non-verbal communication for sexual activity, nor does anything about products for sexuality knowledge of the body. Simply when dealing with TBI, some could have hemi-neglect, there is nothing at the perceptual level.

The Case Description section is very interesting and I think it would improve if the authors improved the information display structure. I think the authors have done a great job at this point, but it is perhaps not very structured and it is difficult to follow the thread.

DISCUSSION

Where is the exchange of information, the contrast of the previous literature with the results of the authors, a lot of work is missing in this section of the document

Round 2

Reviewer 3 Report

Thanks to the authors for the modifications made. Although they have made the appropriate modifications, I believe that the limitations should include the non-inclusion of the occupational therapist and the physiotherapist in the study. Thanks. Very interesting and necessary this type of studies, I hope that the authors continue with this line.
